# Building Pluripotency Identity in the Early Embryo and Derived Stem Cells

**DOI:** 10.3390/cells10082049

**Published:** 2021-08-10

**Authors:** Paola Rebuzzini, Maurizio Zuccotti, Silvia Garagna

**Affiliations:** 1Laboratory of Developmental Biology, Department of Biology and Biotechnology “Lazzaro Spallanzani”, University of Pavia, Via Ferrata 9, 27100 Pavia, Italy; 2Centre for Health Technologies (CHT), University of Pavia, Via Ferrata 5, 27100 Pavia, Italy

**Keywords:** peri-implantation embryo, EGA, lineage specification, pluripotent stem cells, pluripotency transcriptional networks, epigenetic, DNA methylation and histone modification, X chromosome inactivation, non-coding RNAs

## Abstract

The fusion of two highly differentiated cells, an oocyte with a spermatozoon, gives rise to the zygote, a single totipotent cell, which has the capability to develop into a complete, fully functional organism. Then, as development proceeds, a series of programmed cell divisions occur whereby the arising cells progressively acquire their own cellular and molecular identity, and totipotency narrows until when pluripotency is achieved. The path towards pluripotency involves transcriptome modulation, remodeling of the chromatin epigenetic landscape to which external modulators contribute. Both human and mouse embryos are a source of different types of pluripotent stem cells whose characteristics can be captured and maintained in vitro. The main aim of this review is to address the cellular properties and the molecular signature of the emerging cells during mouse and human early development, highlighting similarities and differences between the two species and between the embryos and their cognate stem cells.

## 1. Introduction

The ability of a cell to differentiate and give rise to different specialized cell types represents the cell potency. Thus, depending on a cell’s differentiation potential, potency spans from totipotency, pluri-, multi-, oligo- or uni-potency [1,2]. The fusion of two highly differentiated cells, an oocyte with a spermatozoon, gives rise to the zygote, a single totipotent cell, which has the capability to develop into an entire, fully functional organism. Then, as development proceeds, totipotency becomes restricted at stages that vary among species.

In the mouse, following the first cell division, the blastomeres of 2-cell stage embryos may not be equally totipotent. In fact, following bisection, the pairs may develop into two live-born mice with variable frequency [3,4,5,6], which cannot be ascribed to detrimental effects of the bisection procedure itself [7]. Rather, unequal segregation of the zygote cytoplasmic components [8] may influence totipotency continuity in the blastomere pairs. Difficulties in preserving blastomeres undamaged hampered the possibility to document reproductive totipotency in 4- and 8-cell mouse embryos (for a review, see [9]). Then, embryonic cells evolve towards pluripotency while the embryos undergo through cellular events that occur at specific time points after fertilization. At 2.5 days *post coitum* (dpc), 8-cell stage embryos undergo compaction, followed by morula cavitation and blastocyst formation (3.5 dpc), the latter constituted of an outer single-layered epithelium, the trophectoderm (TE) and an inner cell mass (ICM) facing a fluid-filled cavity (blastocoel) (Figure 1). While the TE forms the fetal component of the placenta, the ICM, initially made of common progenitor cells [10], gives rise, through a second lineage specification, to the epiblast (EPI) and the primitive endoderm (PrE) (Figure 1) [11]. During this time window, EPI cells become pluripotent, i.e., able to develop into ectoderm, mesoderm and endoderm (the three germ layers), to the germ line and to the extraembryonic ectoderm and mesoderm. At 4.5 dpc, following hatching from the zona pellucida, the mature blastocyst implants into the endometrium and gastrulation will then follow at 6.5 dpc.

During human development, these processes are slightly delayed. Totipotency is typical of the zygote and the 2-cell stage embryo [12], but when and how blastomeres lose totipotency is still debated. Unequal potential of the 4-cell blastomeres was highlighted in a study where 4-cell embryos were dissociated and 67% of blastomeres reached the blastocyst stage [13]. At 4 days post-fertilization (dpf), at the 16-cell morula stage, embryos undergo compaction, intercellular adhesion increases and blastomeres flatten. On day 5, cavitation occurs, leading to the formation of a fluid-filled blastocoel cavity. At this stage, the blastocyst shows a compact ICM surrounded by TE cells. Then EPI and PrE segregation occur at 6 dpf [14] (Figure 1). In human, no ICM specific molecular signature has been identified so far, and thus, only the EPI can be considered pluripotent [15,16].

The embryo cell potency can be captured and transferred in vitro, and both human and mouse peri-implantation embryos are a source of different types of pluripotent stem cells (PSCs), whose characteristics depend on the developmental stage of the embryo. The first lines of stem cells (SCs), named embryonic stem cells (ESCs), were derived from the ICM of mouse 3.5–4 dpc blastocysts in 1981 by Martin [17] and by Evans and Kaufman [18] and, 17 years later, from the ICM of human 5 dpf blastocysts by Thomson and colleagues [19]. In 2007, mouse epiblast stem cells (EpiSCs) were originally derived from the EPI, dissected from 5.5 dpc [20] or 5.75 dpc [21] post-implantation embryos. Following these first studies, EpiSCs were also derived from embryos up to 8 dpc [22]. Although strongly influenced by culture conditions [23], ESCs and EpiSCs represent an extraordinary tool for understanding cell pluripotency, the mechanisms that underlie its identity, maintenance and evolution. These mechanisms include the fine regulation of transcriptional networks, epigenetic landscapes, families of RNAs and several inter-related molecular pathways. 

The main aim of this review is to address the molecular identity of cell potency in early embryos and their derived SCs. To this purpose, we will describe the cellular and molecular features of the cell potency in mouse and human embryos, highlighting similarities and differences between the two species, and of their derived stem cells, comparing the characteristics of the cell potency when in vitro or in vivo. 

**Figure 1 cells-10-02049-f001:**
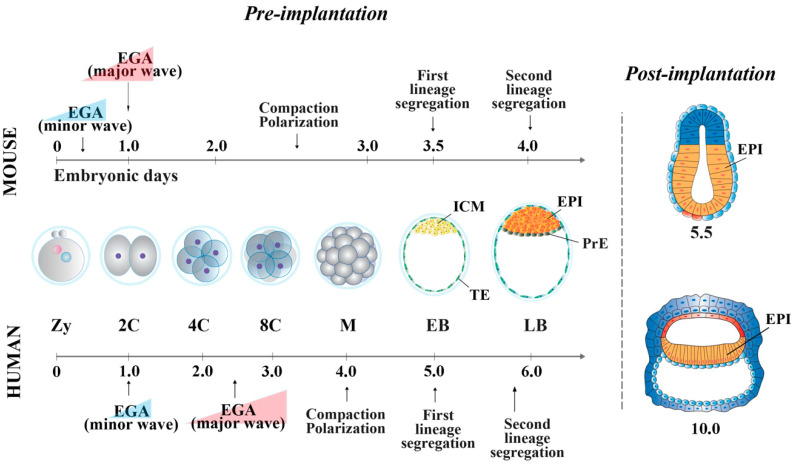
Mouse and human embryonic development from the zygote to early post-implantation stages. Zy, zygote. C, cell. M, morula, EB, early blastocyst. LB, late blastocyst. ICM, inner cell mass. TE, trophectoderm. EPI, epiblast. PrE, primitive endoderm (5.5 dpc and 10 dpf embryos images were redrawn and modified from [24]).

## 2. Cellular and Molecular Features of Mouse and Human Peri-Implantation Embryos

### 2.1. Mouse 

#### 2.1.1. Embryonic Genome Activation

Immediately after fertilization, further development of the totipotent zygote is controlled by maternal-effect factors (RNAs and proteins) accumulated during oocyte maturation [25], which govern the maternal-to-embryo transition and the following embryonic genome activation (EGA). EGA is a mandatory step for the synthesis of new transcripts necessary to the acquisition of an embryonic control of development [26]. Interestingly, Oct-4 (master gene of the pluripotency network; see below) exerts a central role in the maternal-to-zygotic transition [27], and it is expressed in all cell embryos throughout the morula stage, becoming then restricted to cells of the ICM in the blastocyst stage [28].

In the mouse, a minor wave of EGA starts in the zygote at the S/G2 phase, followed by a major wave at the 2-cell stage [26] (Figure 1). When genome transcription begins, major satellites are massively activated, together with transposable elements (TE). These latter comprise long interspersed nuclear element 1 (LINE-1) [29,30], intracisternal A-particles (IAPs) [31,32] and murine endogenous retroviruses with leucine tRNA primers (MuERV-L) [33]. TE transcripts, with different expression kinetics, constitute a significant portion of the transcriptome during mouse EGA. 

EGA coincides with the gradual and progressive degradation of the maternal transcripts. Several RNA-binding protein complexes regulate maternal mRNA silencing to promote their degradation via cleavage, de-adenylation and elimination of their protective cap [34]. The early expression of zygotic microRNAs is necessary for the degradation of hundreds of maternal transcripts [26,35,36]. In addition, specific transcription factors, e.g., the general transcription factor TATA-binding protein, part of the TFIID complex, promote the formation of the RNA polymerase II pre-initiation complex [37] directing the transcriptional machinery during EGA [26]. 

The synergy between EGA and RNA degradation induces genome remodeling, mediated by histone acetylation and methylation, the establishment of topologically associating domains and nucleosome positioning, mandatory for the acquisition of the cellular identity in the embryo [26]. After EGA, molecular differences arise among blastomeres.

#### 2.1.2. Embryo Compaction and First Cell Lineage Specification

At the 8-cell stage, embryos undergo polarization and compaction (Figure 1). Cell polarization is induced by the assembly of the core cell polarity complexes on the outer apical membrane of the blastomeres and by the localization of MAP/microtubule affinity-regulating kinase 2, scribbled homolog and lethal giant larvae homolog 1 [38,39,40] localized on the basal lateral membrane of each blastomere, generating an apical–basal axis [40]. Concomitant with polarization, the expression and localization of E-cadherin and several cytoskeleton and cell adhesion/junction-related proteins allow embryo compaction [41]. At this stage, a switch from all symmetrical to combined symmetrical and asymmetrical cell divisions generates a morula embryo with outside polar and inside apolar cells [40,42].

From the morula stage, lineage specification and differentiation are accompanied by a decrease of global cell potency, the latter achieved through a precise spatio-temporal activation of key genes [43,44]. At this stage, the first lineage specification is determined by several regulatory pathways and, among these, the Hippo signaling pathway has a major role in triggering the specification of the ICM and TE [45,46,47]. The Hippo pathway is selectively inactive in the outside cells and active in the inside cells of the embryo [47,48,49]. In outer cells, Yap1 and its related protein Wwtr1 (hereafter called Yap) translocate and accumulate in the nucleus, where they activate the TE-specific transcriptional program. Specifically, Yap1 promotes the expression of *Cdx2* and *Gata-3*, which together with *Notch*, *Eomes* and *Elf5* contribute to its differentiation [14,50,51], induce TE fate [52,53,54,55,56,57], while repressing the expression of *Sox2* [58,59]. In the inner cells, Yap1 is phosphorylated and retained in the cytoplasm, where it is degraded; thus, the Hippo pathway is activated, allowing transcription of ICM-specific genes (e.g., *Oct-4*, *Nanog*, *Sox2* and *Esrrb*) [60]. Mouse ICM is composed by heterogeneous cells, which co-express different levels of lineage-associated factors, such as Gata-6 (PrE) and Nanog (EPI) [61,62].

#### 2.1.3. Second Cell Lineage Specification

At the blastocyst stage, cells of the ICM undergo a second lineage specification determined by the FGF/FGF-receptor signaling, mediated via MEK/ERK [63,64]. Gata-6 and FGF/ERK induce PrE fate [53,65,66], whereas Gata-4 and Sox17 are involved in its maintenance [65,67]. EPI cells are characterized by the expression of *Oct-4*, *Nanog*, *Sox2*, *Kfl2*, *Klf17* and *Esrrb*, all involved in pluripotency maintenance [14,51,53,68]. Specifically, the establishment and maintenance of the pluripotency state is characterized by a well-defined “pluripotency gene regulatory network” (PGRN) and by a strong cooperation of transcription and epigenetic factors that act in synergy [69,70]. The PGRN is a very highly interconnected system, in which Oct-4, Nanog and Sox2 represent the central functional core. Among this triad of genes, *Oct-4* is at the top of the pluripotency regulatory hierarchy, being essential to reach and maintain pluripotency [71,72,73]. *Oct-4*, *Nanog* and *Sox2* function together, regulating their own promoters and forming an auto-regulatory loop [74,75,76]. Several transcription factors enhance the auto-regulation of Oct-4 and Sox2, either directly or indirectly via Nanog [77]. Specifically, Oct-4 interacts also with Sall4 [78], Zfp322a, Egr1 [79], Utf1 [80] and Dpp4a [81]. Similarly, Nanog interacts Sall4 [78], STAT3 [82], c-fos [83], Utf1 [80] and Dpp4a [81]. Sox2 interacts with Sall4 [78], Utf1 [80] and Dpp4a [81].

The Oct-4, Nanog and Sox2 functional core is involved in the formation of multiple-gene networks that govern cell pluripotency, enhancing the genes necessary to maintain the pluripotent status and repressing transcription, in a target gene-dependent manner, of genes encoding differentiation signals.

### 2.2. Human

#### 2.2.1. Embryonic Genome Activation

Human embryos show two waves of EGA, a minor wave, at the 2-cell stage [84,85], and a major wave, between the 4- and 8-cell stage [26,86] (Figure 1). EGA coincides with the gradual degradation of the maternal transcripts and, as reported for the mouse, TE (SINE-VNTR-Alus [87]; HERV [88]) are heavily transcribed. 

#### 2.2.2. Embryo Compaction and First Cell Lineage Specification

Studies are still needed to clarify the sequence of molecular events that regulate the human first and second lineage specification. Polarization, which initiates at the 8-cell stage, and compaction lead to the morula stage (Figure 1). The molecular mechanisms that drive these processes in human are still unknown, but as they are highly conserved in all mammalian species (except for the timing), it has been hypothesized that human blastomere compaction is driven by actomyosin cytoskeleton and E-cadherin as in other species [89]. By the time embryos reach compaction, apical microvilli and basolateral E-cadherin expression have been observed [90,91].

At the compacted morula stage (4 dpf), both specific TE determinants and ICM-related genes are expressed to determine the first lineage specification [66]. The lineage-specific transcripts become mutually exclusive only at the early blastocyst stage (5 dpf). However, it has been shown that 5 dpf TE cells still retain the ability to form ICM cells [92], and, conversely, isolated ICMs can also generate TE cells [93], indicating that cells at this stage of development are not yet fully committed. In human embryos, the specification of the cell lineages does not seem to occur through a stepwise process, as for the mouse [66], with transcriptional differences being detected at 5 dpf, once the blastocyst is formed. Additionally, it is unclear whether Hippo pathway determines the first cell fate decision, as, in early blastocyst, YAP1 is present in the nucleus of all blastomeres, but, in the late blastocyst at 6 dpf, its expression becomes restricted only to TE cells [55,94]. GATA-3 and CDX2 are also involved in TE cell determination [53,56,95]. 

#### 2.2.3. Second Cell Lineage Specification

Human EPI cells display the expression of *NANOG*, *OCT-4* and *SOX2*, all required for pluripotency maintenance, and of *KLF17* [53,68,96]. GATA-6, initially broadly expressed in the early blastocysts, is involved in PrE fate induction [56,65], together with SOX17 and GATA-4. These latter two are expressed later and are restricted to the PrE, where they are required for its maintenance [53,65]. Very recently, successful 3D culture systems, able to sustain embryos from 5–6 up to 14 dpf, confirmed the results reported above and allowed a finer tuning of the molecular characteristics of the early blastocyst cells and their derived cell lineages [97,98]. At 5–6 dpf, almost all blastomeres show GATA-6 expression, whereas OCT-4, NANOG, KLF17 and PRDM14 are expressed only in the ICM cells. At 7–8 dpf, following lineage specification, PrE, TE and EPI express GATA-6, CK7 and OCT-4, respectively, and at 9 dpf, the three cell lineages acquire a more specific molecular identity and EPI-specific genes are associated with signaling pathways involved in the regulation of stem-cell pluripotency; among these, PI3K–AKT, p53, RAP1 and MAPK are the best defined [98].

In summary, both mouse and human pre-implantation embryos display two consecutive waves of transcription, which determine the activation of the embryonic genome. These waves occur at different stages of development, as well as cell polarization and compaction. The majority of genes that mark the EPI, the PrE and the TE are shared between the two species, with the exception of Eomes, involved in the TE fate, and Esrrb and Klf2, involved in the EPI fate, only in mouse embryos. In this same species, the Hippo pathway governs the first cell fate decision, whereas the FGF/ERK pathways are necessary for PrE induction in both species.

## 3. Features Governing Cell Potency

### 3.1. The Epigenetic Landscape

During the early phases of development, molecular modifications, e.g., DNA methylation, histone methylation and acetylation, and, for female embryos, the progressive silencing of X-linked genes for dosage compensation, determine specific epigenetic landscapes, which contribute to the progressive acquisition of pluripotency. 

#### 3.1.1. DNA Methylation and Histone Modifications

##### Mouse

After fertilization, the female and male genomes undergo genome-wide epigenetic reprogramming, which induces a reset of the gametes’ epigenetic profiles to a basal state, with the exclusion of imprinted domains (ID) and some classes of repetitive elements (RE). ID and RE are protected from demethylation by the four maternal-effect genes SETDB1, ZFP57, TRIM28 and DNMT1 [99,100] (Figure 2). 

Within 4–6 h after zygote formation, the mouse male genome undergoes an active widespread erasure of DNA methylation [101,102], achieved through Tet enzymes (in particular TET3 methylcytosine hydroxylase) and only to a limited extent by base excision repair components [103] (Figure 2). 

During the protamine/histone replacement period, the conformation of the naked male genome permits the accessibility to demethylases, favored by the affinity for specific regions of binding and by the low packaging of nucleosomes for the presence of acetylated histones [104]. In parallel, H3K7me3 accumulates at the peri-centromeric regions, initiating their silencing [105], whereas H3K9me3 [99] protect from DNA methylation erasure. Then, histone acetyl groups are replaced by monomethyl groups and by H3K4me1, H3K9me1 and H3K27me1 activator histones [106,107]. The levels of histone arginine methylation (on H3 and H4 histones) and lysine acetyl modification, markers of active gene expression, significantly decrease [108,109]. On the contrary, the mouse female genome maintains its DNA methylation profile unaltered up to the 4-cell stage, since it escapes the active demethylation process. The female factor Dppa3, by interacting with H3K9me2 (enriched in the female, but not in the male pronucleus), concurs with the reduction of TET3 affinity and protection of 5-methylcytosines from oxidation [110,111] (Figure 2). In addition, acetylated lysines and methylated histones H3K4me1, H3K4me3, H3K9me2/3, H3K27me1, H4K20me3, H3K27me3 and H3K64me3 are the typical epigenetic signature of the female genome [109], rapidly erased after fertilization and re-established later only at CpG islands and active promoters [112,113] (Figure 3). Repressive H3K64me3 and H4K20me3 gradually decrease after fertilization [114,115], whereas repressive H3K9me3 and H3K9me2 are maintained at centromeric major and minor satellites, respectively [106,116,117] (Figure 3). The chromatin state of pre-EGA embryos is in an open conformation and the regions where EGA starts are marked by H2Aac, H3ac and H4ac histone modifications [118] (Figure 3).

In the mouse, SINEs and LINEs RE gain methylation during the transition from the morula to the blastocyst stage [119,120]. Thus, blastocyst formation and the cell commitment to form ICM or TE is flanked by a second wave of asymmetric epigenetic remodeling (Figure 3). Once lineage specification begins, morphological changes are marked by new epigenetic signatures. Although the epigenetic regulation during lineage specification is not fully characterized, a de novo acquisition of repressive histone markers (H3K9me2 and H3K27me3) is mandatory to guarantee the proper lineage specification and the transition towards pluripotency [121,122]. In addition, some regulatory domains and promoters in the ICM show both the presence of H3K4me3, an activation-associated modification, and H3K27me3, a repression-associated modification. Activating and repressive marks localize at the same genomic portion, determining a characteristic chromatin status called bivalent. This status maintains developmental genes expressed at very low levels, although, at the same time, it keeps them poised and ready to be activated; the massive establishment of this chromatin regulation modality is observed during the transition from the morula to the blastocyst stage, when the first lineage specification (ICM and TE) occurs [123]. Additionally, ICM cells are enriched of repressive H3K27me1/2/3 and activating H3K9ac, but also of H4K8ac, H4K16ac, H3K27ac and H3K4me3, found at the *Oct-4* and *Nanog* regulatory regions. Similarly, in TE cells, *Cdx2* promoter is enriched of the same active H4K8ac and H3K4me3 markers [124,125]. As soon as the lineage specification moves on, many bivalent domains are resolved to a monovalent mark (either H3K4me3 or H3K27me3) [123], inducing either gene expression or repression [126,127]. When ICM cells receive differentiation signals, H3K27me3 histone variant is removed from the lineage-specific differentiation genes, while it is maintained in developmental control genes not relevant to the specific commitment [128]. Bivalent domains also exist in pluripotent EPI cells of early post-implantation embryos. However, in lineage-restricted trophoblast and extraembryonic endoderm stem cells, there are few of these bivalent domains and repressive H3K27me3 is replaced by H3K9me3 [129].

##### Human

In human zygotes, the male genome, which displays a lower DNA methylation level than the female one, is actively and rapidly demethylated, whereas the female genome maintains its oocyte-derived DNA methylation pattern up to the third cell division [130,131,132,133,134]. Then, from morula to the blastocyst stage, a second reduction of DNA methylation level occurs [133]. Before the major wave of EGA, at the 4-cell stage, the activator histone variant H3K4me3 is localized at the promoters of several genes and 53% of this mark remains up to the 8-cell stage. The sites in which H3K4me3 is lost (~47%) are the promoters of genes related to development and differentiation, which remain inactive during EGA. H3K27me3, highly present in GV oocytes, is absent in 8-cell stage embryos, indicating its global erasure from the female genome [135,136]. In embryos, the lack of this histone has been correlated with absence of the core components of polycomb repression complex 2 (PRC2) [137] and of an imprinting regulation (e.g., XCI), as reported for the mouse [66,138].

As for the mouse, de novo methylation occurs at the blastocyst stage in human embryos [139]. When cultivated in vitro up to 14 dpf, DNA methylation levels highly increase, with distinct and asynchronous patterns, from day 6 to day 10 dpf in all the three EPI, TE and PrE lineages [97].

In synthesis, in both mouse and human embryos, the male genome is actively demethylated before the first cell division, whereas the female genome is passively demethylated during the following divisions. Then, in both species, de novo methylation occurs at the blastocyst stage. At each stage of development, together with DNA methylation, several distinct acetylated and methylated histones progressively concur in shaping the embryonic epigenome, which, mainly during the transition from morula to blastocyst, acquires a chromatin bivalent status in the mouse, but that has never been described in human embryos.

#### 3.1.2. X Chromosome Inactivation and Reactivation

During pre-implantation development, the establishment of the totipotency condition and the following transition towards pluripotency is accompanied by another important epigenetic event, which occurs only in the female pre-implantation embryos, i.e., the X chromosome inactivation (XCI). Although with different dynamics in mouse and human [140], the XCI compensatory mechanism is mediated by the expression of the long non-coding RNA (lncRNA) Xist (see Section 3.2.2 Long non-coding RNAs), through which one of the two female X chromosomes (paternal X (Xp); maternal X (Xm)) is randomly inactivated to equalize X-linked gene expression between male and female individuals [141]. Opposite to XCI, reactivation of the Xi chromosome, leading to an Xa, occurs only in the mouse female embryos.

##### Mouse

In the mouse, X inactivation occurs through two subsequent waves, the first at the 2-/4-cell stages and the second after blastocyst implantation. After EGA, Xist selectively coats and silences the Xp (imprinted XCI), maintaining this condition up to the morula stage and then in TE cells. This event is also accompanied by the accumulation of the repressive H3K27me3, H3K9me3 and PRC2 chromatin remodeling complexes, together with the loss of activating H3K4 methylation and H3K9 acetylation, the inclusion of macro-H2A and an extensive DNA methylation [142]. Prior to the second wave of random XCI, in mouse EPI cells, X chromosome reactivation (XCR) occurs [143,144]. This process, opposite to XCI, consists of the reactivation of the inactive Xp (Xip) to become an active X (Xap). It gradually takes place through three distinct phases (initiation, progression and completion), through which the epigenetic memory is progressively erased, followed by transcriptional gene reactivation and X chromosome-wide chromatin remodeling [145,146]. Although with a mechanism that is not completely known, mouse Xip reactivation takes place within few hours [144] and starts before the commitment of PrE and EPI lineages. It occurs in some ICM cells before Xist downregulation and H3K27me3 loss, suggesting that the process is independent from Xist silencing [144,147]. Progressively, the biallelic gene reactivation becomes restricted to the pre-EPI cells and strongly correlates with Xist silencing, Tsix expression (negative regulator of Xist), loss of the epigenetic memory and the expression of Nanog protein [143,144]. In addition, the enrichment of chromatin marks associated with gene repression (e.g., H3K27me3), macroH2A and DNA methylation are progressively removed [143,144,148,149,150]. Upon blastocyst implantation, mouse EPI cells undergo random XCI and either the Xp or the Xm is subject to inactivation, with a stochastic choice that appears to be made independently in each cell [151], after which the inactive X is clonally inherited to the cell progeny.

##### Human

Differently from the mouse, in human pre-implantation embryos, the precise mechanism of X dosage compensation remains still under discussion [152]. A current model proposes that X chromosome dampening (XCD) [66], i.e., a progressive decrease of biallelic expression of X-linked genes, occurs shortly after EGA and continues throughout pre-implantation [66,97,153]. By the time of implantation, either the Xp or the Xm is inactivated, followed by an upregulation of Xa-linked genes [66,97,154]. With the exception of the female germ line, where the Xi is reactivated, the XCI choice made at the time of implantation is clonally inherited throughout development in the somatic lineage.

In summary, Xist, cooperating with several other lncRNAs, governs the X dosage compensation in both mouse and human pre-implantation embryos. However, the events through which the inactivation of the X chromosome occurs in the two species are different and, for humans, the precise mechanism of X dosage compensation still remains debated.

### 3.2. Non-Coding RNAs

Non-coding RNAs are an abundant heterogenous group of RNAs which exert fundamental regulatory functions in mammalian cells. By interacting with DNA, RNA or with proteins, non-coding RNAs are engaged in the modulation of chromatin structure and function, and in the regulation of gene expression, RNA splicing and protein translation. 

During development, they act as modulators of the cell potency and, as miRNAs and lncRNAs are the most abundant in pre-implantation embryos, we will focus our attention on these two families [155].

#### 3.2.1. miRNAs

The expression of specific miRNAs at specific embryonic stages is relevant for a correct embryonic mouse development [156] and characteristic miRNA profiles were identified at each pre-implantation stage [157,158]. 

The fully-grown oocyte and the zygote have a very similar miRNA profile, as the majority of zygotic miRNAs are maternally inherited [156,159]. Besides the maternal inheritance, at the time of fertilization, the sperm delivers several small RNA species to the oocyte, including miRNAs [160], whose role, although not yet clearly understood, is critical to embryo development, since their lack results in embryonic failure. For example, sperm-borne miR-43c is required for the initiation of the first cell division [161]. Between the zygote and the 2-cell stage, a significant global loss of maternal miRNAs occurs: about 60% of the miRNAs pool was downregulated and some miRNA were reduced by 95%.A high-throughput profiling of miRNAs has evidenced that 3′ mono- and oligo-adenylation modifications frequently occur in the zygote and in the 2-cell embryos. This modification protects miRNAs from degradation and represses their function during these two phases, although they may be re-activated during the following cleavage stages [162]. At the 2-cell stage, after EGA, a de novo synthesis of miRNAs starts and, among these, miR-290 to miR-295 are the first to be detectable [159]. Precursors miR-20a and miR-292 are also observed, followed by their mature forms in the subsequent 4-cell embryos [163]. The progression from the 4-cell to late blastocyst stage is marked by reduced capacity for miRNAs processing, as demonstrated by the progressive downregulation of the genes coding for proteins involved in their biogenesis [163]. Interestingly, the expression of the miR-290-295 cluster, which has an upward trend from the zygote to the blastocyst stage, is directly controlled by Oct-4 and Nanog [164], which, together with Sox2, constitute the core of the PGRN. At the blastocyst stage, in EPI cells, members of the miR-290 cluster, including miR-292-3p and miR-292-5p, are highly expressed and a significant increase in miR-292-3p and miR-292-5p is reported in the latest stages of pre-implantation development [163].

The meagre knowledge of miRNAs’ role during human development arises from oocytes or embryos used for Assisted Reproductive Technology (ART) procedures, in which specific miRNAs were identified as biomarkers of human oocytes’ competence, embryos’ quality or of successful implantation outcome, without an accurate characterization of their functions. For example, increased expression of hsa-miR-142-3p and decreased expression of hsa-miR-20a and hsa-miR-30c have been identified in the culture media of non-implanted blastocysts, when compared to that of implanted embryos [165,166]. Similarly, miR-320a and miR-15a-5p were found highly abundant, whereas miR-21-5p and miR-20a-5p were significantly less abundant, in the media where low-quality embryos were cultured, when compared to their presence in the culture media of excellent quality embryos [167]. To the best of our knowledge, no studies focus on miRNAs and their relationship with the acquisition of pluripotency during human embryonic development.

#### 3.2.2. Long Non-Coding RNAs

LncRNAs are part of the dynamic transcriptional variations that parallel morphological changes occurring during mouse and human early embryo development [168]. LncRNA abundance changes at each stage of development; for example, in the mouse, they represent 42.1% of all the zygote’s RNAs, reaching 53.2% in the blastocyst, suggesting that they become predominant over protein-coding RNAs at the late stages of pre-implantation development [169]. LncRNAs are expressed in a temporal-specific manner, more in mouse than in human early embryos [170], and they are also heterogeneously expressed in seemingly identical cells [171], suggesting their contribution in the acquisition of cell identity and in cell potency changes.

Within the lncRNA family, hereafter we briefly report on XIST and XACT, chosen for their role in the XCI process, and on LincGET for its role as an early regulator to bias cell fate in mouse 2-cell embryos [172].

Historically, the first identified lncRNA was XIST, an untranslated spliced 17-kb-long molecule, which triggers *cis*-inactivation of the X chromosome during the early human developmental phases. It is transcribed at a low level from both Xa chromosomes, and then, it is upregulated and expressed from the presumptive Xi [173,174], forming a cloud-like structure in the nucleus and leading to gradual specific X-chromosome silencing. To mediate this process, XIST acts as a platform for the recruitment of the polycomb repressive complex 1 (PRC1) and PRC2 chromatin remodelers, histone deacetylases, histone variants and the entire DNA methylation machinery [175,176,177,178], modifying the chromatin organization of the decorated X-chromosome and its positioning within the nucleus [179,180,181,182]. In human embryos, a 252 kb lncRNA, named *XACT* (X-active coating transcript) [183], participates in the compensatory mechanism occurring during the early stages of development [66]. It co-accumulates with *XIST*, controlling the association of XIST to the putative Xi *in cis*, possibly to antagonize or temper its silencing ability [184]. In mouse and human peri-implantation embryos, several other lncRNAs (e.g., *Tsix*, *Jpx*, *Xite*, *Ftx* and *Tsx*, for mouse; *TSIX*, *JPX* and *FTX*, for human) concur with *XIST* in the silencing process [140]. 

Very recently, an endogenous retrovirus (ERV)-associated lncRNA, called LincGET, with an essential role in embryo development has been identified [185]. It is expressed during the early mouse developmental phases, along with EGA, appearing first at the early 2-cell stage. It is upregulated through the late 2- to early 4-cell stage, downregulated through the late 4- to early 8-cell stage and subsequently undetectable at the late 8-cell stage. LincGET expression was highly heterogeneous in single blastomeres, specifically in 4-cell blastomeres compared to that in 2-cell embryos, suggesting a role in directing the developmental fates of early blastomeres. Its heterogeneity at the 4-cell stage correlates with the expression of CARM1, a protein arginine methyltransferase, which accumulates in nuclear granules in the 2- to 4-cell stage embryo and is responsible for the histone H3R26me2 modification heterogeneity in 4-cell embryos [186]. Through its interaction with CARM1, LincGET controls alternative splicing [185] and regulates gene expression [172].

## 4. Cellular and Molecular Features of Stem Cell Pluripotency In Vitro

SCs, isolated either from embryos at different stages of pre- and early post-implantation development or from PSC lines, can be maintained in vitro applying self-renewal culture conditions [187,188] and possess cellular and molecular features that mirror different pluripotency states, defined as extended, naïve, intermediate and primed. They are characterized by distinguishable colony morphology, growth factor requirement, energetic metabolism, molecular signatures and, in the female cell lines, X inactivation status [70,189,190]. 

### 4.1. Cellular Features

#### 4.1.1. Mouse

##### Stem Cells with Naïve Pluripotency

Mouse ESCs retain the same molecular and transcriptional features of the EPI cells present at 4.5 dpc pre-implantation embryo stage [191], with a pluripotency characteristic called “naïve” [189] (Figure 4). In vitro, naïve mESCs grow as small, compact and domed-shape colonies, and they display high clonogenicity capability. Using leukemia inhibitory factor (LIF), mESCs can be propagated without feeder cells on gelatin-coated plates. Serum/LIF is the standard culture condition that allows the maintenance of naïve mESCs potency, suitable for blastocyst chimera formation. Indeed, when injected in early pre-implantation embryos, naïve mESCs contribute to all somatic lineages and to germline, indicative of their pluripotency in vivo [192]. Mouse ESCs are in an unstable balance between pluripotency and differentiation signals, which support self-renewal (maintained by LIF) or promote differentiation (induced by FGF). However, the addition of exogenous LIF favors self-renewal at the expense of differentiation [193].

As ICM cells from which they derive, ESCs are heterogeneous and mainly consist of two cell types, i.e., the primed progenitors of the EPI and PrE [194,195,196,197]. In vivo, bipotent progenitors exist for a short transient time window prior to implantation, as they rapidly become committed [10]. On the contrary, in vitro, these two cell states are maintained indefinitely and dynamically interconvert in Serum/LIF culture conditions [195] and the regulation of this interconversion involves the activity of PRCs [198]. Additionally, mESC lines exhibit a small subpopulation (about 1%), called 2-cell like (2C-like), with several features of the 2-cell embryo blastomeres [33,199,200], such as the expression of specific genes and repeats (e.g., *Zscan4* genes and MuERV-L repeats) [201], dispersed chromocenters and high histone mobility [33,202,203]. Additionally, when injected into morula stage embryos, 2C-like cells contribute to both embryonic and extraembryonic tissues [33]. In serum/LIF conditions, these 2C-like cells can spontaneously derive from mESCs and multiple factors, such as Tet proteins [204], miR-34a [205] and components of noncanonical PRC1 [206], which concur in the pluripotent-to-2C-like state transition. The appearance of this subpopulation is due to a two-step reprogramming process, which entails, at the beginning, the downregulation of pluripotency genes (e.g., *Oct-4*, *Sox2*), followed by the activation of the expression of 2-cell-specific transcripts (e.g., *Zscan4*) [207]. 

More recently, the dual inhibition (2i) culture system, a defined serum-free medium (N2B27) with small molecule inhibitors of the MAPK/ERK pathway (PD0032, a MEK inhibitor) and of the Glycogen Synthase Kinase 3 (GSK3) (CHIR99021 or CHIR), ameliorated the maintenance and the propagation of mESCs [208,209]. Dual inhibition of MEK1/2 and GSK3, optionally in combination with LIF (2i/LIF medium), allows mESCs to maintain the transcription profile, DNA hypomethylation status and developmental potential characteristic of the pre-implantation EPI, a condition referred to as “naïve ground state” [191,210,211,212,213,214]. Following 2i culture, mESCs do not respond to differentiation signals directly and they necessitate a capacitation passage, prior to engage differentiation towards the three germ layers and primordial germ cells (PGCs). ESCs progression from the naïve ground state, initiated by the removal of the inhibitors of the MAPK/ERK pathway and of GSK3 [215], undergoes reprogramming of the pluripotency transcription factor networks, a metabolic reorganization, epigenome and chromatin remodeling, conferring to them the ability to differentiate [122,215,216,217,218,219,220]. The complete sequences of events and molecular mechanisms that accompany this passage are not characterized [215].

##### Stem Cells with Intermediate Pluripotency 

In recent years, SCs, derived from either 5–6.5 dpc mouse embryos or naïve ESCs, have been shown to possess intermediate states of pluripotency between naïve and primed (see below). These transition pluripotency states are defined as “poised” [221], “rosette” [222] or “formative” [223,224,225,226] (Figure 4). These cells, while downregulating the naïve transcriptional program, begin to acquire the competence for multi-lineage differentiation, although they do not yet express lineage-associated markers.

Poised pluripotency cells express high levels of most pluripotency transcription factors (TFs) do not express markers of primed pluripotency, and are characterized by expression of a specific set of mRNAs and miRNAs. Poised pluripotency is, thus, considered as an intermediate phase that precedes the formative status [221].

The recently described rosette pluripotency [222] is characteristic of cells that exist in 5 dpc embryos and in ESCs cultured in conditions where both WNT and FGF/ERK signaling are inhibited. Rosette PSCs co-express the naïve marker Klf4 and the primed marker OTX2. The transition from naïve to rosette pluripotency is guided by the downregulation of WNT signals, whereas the activation of the MEK pathway induces the progression towards primed pluripotency [222]. 

The pluripotency states captured from 5.5–6.5 dpc mouse embryos have been called formative. Formative PSCs display a unique transcriptome and gene regulatory networks, which comprise signaling pathways and epigenetic machinery necessary to acquire competence for lineage specification [223,224,225,226] (see Section 4.1 and Section 4.2).

##### Stem Cells with Primed Pluripotency

Mouse EpiSCs, derived from the EPI of post-implantation embryos at 6.25–8 dpc (Figure 4), are characterized by a pluripotency status named “primed”. mEpiSC colonies grow as a flatten monolayer [20] and require Activin and FGF signaling for their maintenance in vitro [20,21]. As naïve mESCs, primed mEpiSCs display unlimited potential to self-renewal and differentiate into the three germ layers in vitro, but they are limited in their pluripotency in vivo, as they cannot give rise to blastocyst chimeras [20,227,228].

#### 4.1.2. Human

The first derived human ESC (hESC) lines showed primed pluripotency [20,229,230,231,232]. They grow as flatten monolayer colonies, display XCI and poor survival after single-cell disaggregation. hESCs primed pluripotency is maintained when fibroblast growth factor 2 (FGF2) and transforming growth factors β (TGF-β) are present in the culture medium, without supplemented LIF [233,234]. This pluripotency characteristic was first attributed to unspecified genetic differences between mouse and human, but further experiments with ‘non-permissive’ mouse strains demonstrated an in vitro cell adaptation into a primed state during isolation, suggesting the need of culture strain-specific requirements [235].

Many efforts were then made to obtain hESCs with naïve pluripotency. In a series of studies, naïve pluripotency has been successfully acquired from the conversion of primed hPSCs [236,237], by inducing the over-expression of KLF2, KLF4 and OCT-4 pluripotent factors or by adding specific chemicals (e.g., Erk inhibitor PD0325901, Gsk3 inhibitor CHIR99021 and adenylylcyclase activator Forskolin) in the culture medium [236]. Primed cells can also be reverted to the naïve state when cultivated in a 5iLAF medium containing LIF, Activin and/or Fibroblast Growth Factor 2 and a cocktail of five inhibitors, which target MEK, B-Raf, GSK3β, Src and ROCK [238]. 

In other studies, naïve pluripotency was captured from embryos and maintained in vitro following the development of specific derivation and culture protocols [237,238,239,240,241,242]. For example, short-term induction of KLF2 and NANOG allow the derivation of naïve-like hESCs, which are then maintained in medium comprising titrated inhibition of GSK3 and block of the mitogen-activated protein kinase (MAPK/Erk) pathway (t2i) with LIF and protein kinase C (PKC) inhibitor (t2iL+Gö medium) [237]. 

#### 4.1.3. Mouse and Human Stem Cells with Extended Pluripotency

In 2017, a new type of mouse PSCs, defined extended (or expanded) pluripotent stem (EPS) cells, has been derived from 4- or 8-cell mouse embryos, with an efficiency of 20% in feeder-free cultures, and up to 100% on feeder cells [243,244]. EPS cells express pluripotency genes similar to naïve mESCs, display normal karyotype, form teratomas and contribute to both somatic and germline lineages in chimaeras. Once injected into morulae, EPS cells contributed both to the ICM and to the TE, generating both embryonic and extra-embryonic lineages in vivo [243,244]. Two years later, under similar in vitro culture conditions (medium supplemented with inhibitors for GSK3 (CHIR99021), SRC (WH-4-023) and Tankyrases (XAV939), Vitamin C, ACTIVIN A and LIF), human EPS cells were obtained from established hESC lines. These cells possess expanded potency for both embryonic and extra-embryonic cell lineages in vitro [245]. 

In synthesis, mouse naïve PSCs, derived from 3.5–4 embryos, show the same molecular and transcriptional features of the EPI cells present at 4.5 dpc pre-implantation embryo stage. As the cells from which they derive, naïve PSCs are heterogeneous and mainly consist of two cell types (the primed progenitors of the EPI and PrE). In vitro, these two cell states are maintained indefinitely and dynamically interconvert. Additionally, when injected in early pre-implantation embryos, they contribute to all somatic lineages and to the germline. All these features can be influenced by the culture conditions in which they are derived and maintained. PSCs with intermediate states (poised, rosette and formative) between naïve and primed pluripotency have been isolated from either 5–6.5 dpc mouse embryos or naïve ESCs and begin to acquire the competence for multi-lineage differentiation. Primed PSCs, derived from derived from the EPI of post-implantation mouse embryos at 6.25–8 dpc, display unlimited potential to self-renewal and differentiate into the three germ layers in vitro, but they cannot give rise to blastocyst chimeras.

The first derived human ESC lines showed primed pluripotency. Naïve pluripotency can be obtained from the conversion of primed hPSCs or can be captured and maintained from embryos applying specific culture protocols.

Mouse and human extended PSCs, derived from 4- or 8-cell mouse embryos or hESC lines, express pluripotency genes similar to naïve SCs and, when injected into morulae, they contributed to both embryonic and extra-embryonic lineages. 

For both species, the improvement of the derivation and culture conditions permitted the successful capture and transfer in vitro of the pluripotent continuum features of the developing EPI cells in the form of PSCs. However, culture conditions themselves can influence the pluripotency states of SCs, generating a certain degree of disparity between the in vivo and the in vitro conditions. 

### 4.2. Molecular Features

#### 4.2.1. Pluripotency Transcriptional Networks (PTNs)

##### Mouse PTNs


Naïve


Mouse naïve PSCs express the triad *Oct-4*, *Sox2* and *Nanog* gene core together with a cohort of transcription factors characteristic of pre-implantation EPI cells, such as Klf2, Klf4, Klf5, Stella (Dppa3), Fgf4, Esrrb, Rex1 (Zfp42), Tfcp2l1, Tbx3 and the Alkaline phosphatase activity (Figure 5) [188,246]. While the molecular elements of this transcriptional network are all involved in pluripotency maintenance in vitro, some of these exert a different role during embryo development in vivo. For example, *Esrrb*, a crucial regulatory element of the pluripotency network of mouse and hESCs, is not necessarily required for early EPI development in vivo [247]. Specifically, in vivo, *Esrrb* acts by sustaining the expression of *Cdx2*, *Eomes* and *Sox2*, crucial transcriptional regulators of the TE cell identity. Its expression declines after the early blastocyst stage, becoming prevalent in the TE [248].


Formative


In SCs with formative pluripotency, the entire transcriptome is reorganized, the molecular PGRN governing naïve pluripotency partially dismantled and chromatin accessibility remodeled [226]. Specifically, several genes, e.g., *Abcg2*, *Cldn4*, *Vgll1*, *Gata2*, *Gata3* and *Erp27*, are uniquely upregulated and Nanog transcription appears to be strongly downregulated compared with both SCs possessing naïve or primed pluripotency [249]. The transcriptome and chromatin landscape of formative cells provide the proper molecular signals to induce germ layer formation and germline specification [225,226].


Primed


In mouse pluripotent primed EpiSCs, the pluripotency triad *Oct-4*, *Sox2* and *Nanog* is active, but the expression of naïve state markers *Rex1*, *Stella*, *Klf2* and *Klf4* is low or absent [21]. EpiSCs express the early post-implantation embryo markers, such as Brachyury, Eomes, Gsc, Mixl1 and Fgf8 (mesoderm), Sox17, Gata6, Gata4 and FoxA2 (endoderm) and Oct-6, Nodal, Fgf5, Otx2 and Lefty (ectoderm) [20]. In addition, EpiSCs exhibit higher expression of cell adhesion markers (e.g., Tnc, Col1a1 and Col6a1), TGF-β-, MAPK- and Wnt-associated genes, similar to their in vivo post-implantation EPI counterpart (Figure 5) [250]. During culture, mouse primed EpiSCs are characterized by high propensity of spontaneous differentiation and by the generation of subpopulations. These latter co-express the lineage markers Brachyury and FoxA2 together with Oct-4, Sox2 and Nanog, but the level of expression of mesoderm and endoderm markers is inversely correlated with that of Sox1, a neurectoderm marker, suggesting the presence of at least two subpopulations [251]. EpiSCs display intra- and inter-cell lines heterogeneity in the expression of some of these lineage markers, probably related to different in vitro culture conditions (different growth factors added to culture medium (e.g., FGF2 and activin) or differentially active signaling pathways (FGF, Nodal and Wnt)) [252]. Additionally, a small fraction of EpiSCs exhibits features of naïve pluripotency, such as low Brachyury and low Fgf5 expression, *Oct-4* regulation by its distal enhancer (the *cis*-regulatory element, known to regulate its expression only in naïve pluripotent cells) [217] and high levels of specific naïve pluripotency markers [247]. This subpopulation has gene expression features between those of the naïve and primed pluripotency states [247].

##### Human PTNs


Naïve and Primed


Similar to the mouse, naïve and primed hESCs display some differences in their transcriptional networks [249]. When analyzed with single-cell RNA sequencing, naïve and primed populations present different transcriptomes, but all naïve cell lines analyzed are homogeneous among themselves, as well as primed cell lines. However, a small subpopulation of cells can be identified in the naïve state population that display transcriptional features of primed pluripotency [249].

Beyond the pluripotency triad, naïve hESCs express high levels of GATA-6, KLF4, KLF5, KLF17, DNMT3L, DPPA3, DPPA5, IL6ST and TFCP2L1 [249], whereas primed hESCs express markers such as CD24, OTX2, ZIC2, ZIC3, SFRP2, THY1 and DUSP6 [249] (Figure 5). 

In summary, the *Oct-4*, *Sox2* and *Nanog* triad represents the core of both naïve and primed mouse and human pluripotency networks. The triad is conserved between mouse and human, but few interactor genes are shared between the two species. 

In naïve PSCs, interactor genes, which relate to the triad, are also expressed in EPI cells, while, in primed PSCs, interactors are also found expressed in mesoderm, endoderm and ectoderm germ layers, suggesting that the different pluripotency states are sustained by dissimilar molecular networks, which also change between the two species.

#### 4.2.2. DNA Methylation and Histone Modification

The transfer of embryonic cells from in vivo to in vitro culture entails modifications in the epigenetic landscape that determines their potency. The process of isolation of the ICM from the blastocyst and its cultivation in defined media, suitable for self-renewal maintenance, involves arrest of the normal developmental program, determined by epigenetic regulators which play a key role in this tricky passage. 


Mouse


Naïve mESCs, cultured in 2i/LIF medium, exhibit global DNA hypomethylation, a hallmark of their epigenome [253,254], Dnmt3a/b downregulation (induced by mitochondrial Stat3) and reduced 5mC levels. On the contrary, mESCs cultured with Serum/LIF display higher levels of DNA methylation [255]. Compared to the ICM, mESCs display higher levels of repressive epigenetic marks [79] and several epigenetic regulators have different levels of expression (up- or downregulation), probably to underpin the phenotypic changes occurring during the in vivo/in vitro adaptation. Epigenetic regulators linked to repressive epigenetic status are frequently upregulated and, among these, Dnmt3a, Dnmt3b, Dnmt3l, Mecp2 and Mbd2 show a 2- to 12-fold increase in ESCs compared to ICM cells. In addition, histone deacetylases Hdac5, 6, 7 and 11, H4K20 methyltransferase Suv420h2, H3K9 methyltransferase Ehmt1 and the heterochromatin binding protein Hp1β are significantly increased in ESCs. On the contrary, several epigenetic modifiers associated to active epigenetic status are downregulated. Among these, histone acetyltransferases Ncoa3, Creppb, Kdm4d, H3K27 demethylase Kdm6b and H3K4 methyltransferase Mll3 show a significant decrease, ranging from 2- to 10-fold from ICM outgrowth to ESCs [79]. Thus, globally, ESCs are in a more repressive status compared to ICM cells, probably because these latter need a more epigenetic flexibility, being a transient developmental phase, whereas the repressive epigenetic status reported for ESCs is probably related to culture conditions suitable to maintain them undifferentiated and with a self-renewal capability [23,79]. All together, these features suggest that the current in vitro culture conditions do not accurately reproduce the in vivo environment. The development of the 2i medium permitted to maintain DNA of ESCs into a hypomethylated status comparable to that of in vivo ICM cells [212,213,256]. Indeed, a significant low level of H3K27me3 histone (associated to gene silencing), caused by PRC2, has been described at several promoter regions [211], together with the presence of H3K27me3/H3K4me3 bivalent chromatin, as in ICM cells [123,211]. On the contrary, ESCs cultivated in Serum/LIF medium display similar DNA methylation levels of post-implantation embryos [257,258,259]. 

Similar to naïve mESCs, mouse 2C-like cells show lower global DNA methylation together with low DNMTs expression [33,206].

Compared to ESCs, EpiSCs express different epigenetic regulators and exhibit a close chromatin conformation [260,261]. Histone H3K4me1, a mark of active genes, varies significantly between ESCs and EpiSCs [261]. This histone displays an active role in determining the primed pluripotency state; however, its global redistribution at both enhancers and repressors of lineage determinant factors can induce a spontaneous conversion from EpiSCs to a naïve state [262]. In addition, when compared to ESCs, EpiSCs show a reduced expression of SMARCAD1, a blocker of H2K9me3-mediated heterochromatin formation [263].

Mouse extended PSCs exhibit an intermediate level of 5mC between the naïve and primed ESC states, but higher level of 5hmC [264].


Human


Fewer data are available on the epigenome of naïve and primed hESCs. It has been demonstrated that the conversion from primed hESCs to naïve state using Theunissen [87] or Takashima [237] culture media (see Section 4) determines a global reduction of DNA methylation level, in both CpG (26.9–33.2%) and non-CpG sites (0.19–0.29%), compared to their primed counterparts (CpG, 75.2–81.0%; non-CpG, 0.32–0.60%). The level of DNA methylation of converted naïve hESCs is comparable to that of pre-implantation EPI [87,237]. hESC lines, derived under naïve conditions, display some similarities with the epigenic profile of the human pre-implantation EPI. For example, they showed low levels of DNA methylation, with a marked downregulated expression of DNMT3A and DNMT3B. However, the methylome of naïve hESCs is distinct from the human EPI cells of the embryo, concerning, for example, the DNA methylation at primary imprint sites, which is lost in vitro. Additionally, in naïve hESCs, DNA methylation is uniform across the genome, whereas the pluripotent EPI embryo cells show regions with higher levels of methylation [265].

In summary, naïve mESCs exhibit global DNA hypomethylation, Dnmt3a/b downregulation and reduced 5mC levels. Their DNA methylation level changes when the culture medium conditions change (2i/LIF medium lower methylation level; serum/LIF medium higher methylation level). Compared to the in vivo counterpart, mESCs show higher levels of repressive epigenetic marks and they are globally in a more repressive status, suggesting that the current in vitro culture media do not accurately reproduce the in vivo environment. Primed PSCs exhibit a closer chromatin conformation when compared to naïve cells.

In human, the conversion from primed hESCs to naïve state determines a global reduction of DNA methylation level, which is comparable to that of pre-implantation EPI. Human ESC lines, derived under naïve conditions, display some similarities with the epigenic profile of the human pre-implantation EPI, such as low levels of DNA methylation and marked downregulated expression of DNMT3A and DNMT3B.

#### 4.2.3. X Chromosome Inactivation and Reactivation

Mouse female naïve ESCs and primed EpiSCs exhibit different XCI patterns, which correlate to their differential pluripotency state. Naïve ESCs display two Xa chromosomes, as pluripotent EPI embryo cells [266]; instead, primed EpiSCs have an Xa and an Xi, mediated by Xist expression. During in vitro derivation of EpiSCs, a precise recapitulation of in vivo events occurs, which ends with random inactivation of one of the two X chromosomes, due to Xist expression [140]. Interestingly, the pluripotency gene triad Oct-4, Nanog and Sox2 contribute to the regulation of Xist lncRNA expression in both male and female pluripotent ESCs [76,77,267]. 

Human primed PSCs do not show the dampening of X chromosomes, but three distinct classes of cells with different X chromosome states. Specifically, Class I cells display both Xa chromosomes and low (or undetectable) XIST expression (XaXa); Class II cells show XIST expression, together with H3K27me3 deposition, which cause random inactivation of one of the two X chromosomes (XaXi); Class III cells display downregulated XIST and depletion of H3K27me3 on the Xi chromosome, followed by partial reactivation of a number of Xi-linked genes, generating an X-eroded chromosome (XaXe) [268]. During in vitro culture, a gradual progression through the three classes occurs. 

XCR takes place during the forced conversion of hESCs from primed to a naïve-like state in vitro. During this transition, XIST, expressed in primed cells, is progressively silenced; in parallel, XACT is reactivated [87,183] and H3K27me3 and H3K9me3 repressive histone marks are removed [238,241,269]. These events generate an XIST-negative intermediate condition, in which the Xi is reactivated, giving rise to XaXa cells [154,270]. Progressively, during in vitro culture, reactivation of XIST transcription occurs, generating XIST-positive cells, where the dampening of both X-chromosomes determines the reduction of X-linked gene expression.

In summary, as in vivo, Xist also mediates X chromosome silencing in in vitro cultured PSCs.

Mouse female naïve ESCs and primed EpiSCs exhibit different XCI patterns (XaXa and XaXi, respectively), which is strictly correlated to their different pluripotency state. 

Human primed PSCs do not show the X chromosomes dampening. Culture conditions and the time spent in culture influence the expression of X-linked genes, determining three classes of cells: XaXa cells at the time of PSCs derivation, followed by the establishment of XaXi cells, characterized by random silencing of one of the two X chromosomes, and finally XaXe cells, which display a partial reactivation of X-linked genes of the Xi chromosome. 

#### 4.2.4. Non-Coding RNAs

Non-coding RNAs are regulators of naïve and primed pluripotency states, concurring with the maintenance of self-renewal and the inhibition of multilineage differentiation. miRNAs and lncRNAs represent the best characterized for their role in the modulation of the biological properties of PSCs in vitro [271,272].

##### miRNAs

In ESCs, a single family of miRNAs, possessing the AAGUGC seed sequence, is the most highly expressed [273]. The members of this family are organized in two major clusters, localized on mouse chromosomes 3 and 7, and on human chromosomes 4 and 19. The first cluster, named miR-302/367, is very conserved and comprises miR-302a, miR-302b, miR-302c and miR-302d and the unrelated miR-367 (this latter without the AAGUGC seed sequence). The second cluster, named miR-290-295, is less conserved and includes miR-290, miR-291a, miR-291b, miR-292, miR-294 and miR-295 and miR-293 (without the AAGUGC seed sequence). The human orthologue comprises miR-371 and the AAGUGC seed-containing miR-372 and miR-373 (miR-371-373 cluster) [274]. In both clusters, other miRNAs with the AAGUGC seed are present, as well as other miRNAs with a different seed [275]. miRNAs belonging to both clusters are highly expressed in mouse and human ESCs and their expression rapidly drops upon differentiation [276]. Specifically, in hESCs, the miR-302 cluster represents more than 60% of all expressed miRNAs [277], whereas in mESCs, miRNAs belonging to the miR-290-295 cluster predominates, representing about 30% of their global miRNome [278]. Interestingly, in both mouse and human, genes of the ESC pluripotency core were found to be involved in regulating the expression of miRNAs of these two major clusters. 

NANOG and OCT-4 transcription factors are upstream regulators of the miR-302/367 cluster [279] and, together with SOX2 and TCF3, bind directly to the promoters of miR-290-295 clusters (and of other miRNAs), modulating their expression in both mouse and human ESCs [280]. In association with Polycomb complexes, the same genes repress the transcription of lineage-specific miRNAs (e.g., miRNA-155 mesoderm specific; mir-9/-124 ectoderm specific) [280]. 

miRNAs are key regulators of self-renewal and differentiation of stem cells, as important as transcription factors in controlling gene expression. For example, miR-302 family members were found to be involved in the regulation of the mESCs cell cycle. Specifically, they target the CDK inhibitors p21, Rbl2 and Lats2 in both mESCs [281] and hESCs [282], and in the latter, this family also regulates Cyclins D1 and D2 [283]. Additionally, in hESCs, miR-302 family, miR-145 and miR-296 have a periodic expression along the cell cycle, with an induction at G1/S boundary and high expression levels at S phase [284]. The miR-302 cluster have also a central role in expediting the G1/S transition and promoting cellular proliferation [285]. Instead, miR-145 and miR-296 are induced during differentiation and silence self-renewal [286,287].

The miRNA signature is distinguishable when comparing the naïve with the primed pluripotent states. miRNA markers of human naïve cells are miR-143-3p, -22-3p and the miR-371-373 cluster, the latter very highly expressed. On the contrary, miRNA markers of primed state are miR-363-5p and several members of the miR-17 and miR-302 families [274]. The conversion of mESCs from naïve to the primed condition is shepherded by an immediate upregulation of miRNA-363-3p and miRNA-205-5p, as well as miR-17 family members miR-18b-3p, -20b-5p, -20b-3p and -106a-5p [274]. During this transition, quantitative variations of both miR-290-295 and the miR-302/367 clusters were also detected. The expression of miR-290-295 and miR-142-3p, high in naïve mESCs, drops in EpiSCs, in which miR-302/367 and let7e significantly prevail [273]. Human primed ESCs display higher expression of the miR-302/367 compared to miR-371-373 cluster; the latter takes over during the first steps of differentiation [288].

Transient exposure to miR-203, a non-coding RNA expressed in mouse embryo from the 2-cell to the morula stage, improves the differentiation capacity of both murine and human PSCs. Specifically, short exposure to miR-203 of mPSCs induces a transient expression of 2-cell embryo markers, expanding their differentiation potency to multiple lineages and improving their efficiency in tetraploid complementation and human–mouse chimerism assays. This phenomenon has been associated with a repression of Dnmt3a and Dnmt3b methyltransferases, which induces a transient and reversible erasure of DNA methylation in PSCs [289]. To the contrary, inhibition of miR-34a in mESCs expands their developmental and cell fate potential, becoming capable to differentiate into both embryonic and extraembryonic cell lineages. In miR-34a-deficient PSCs, Gata2 transcription factor increases and MuERV-L are strongly induced, a typical feature of the 2-cell blastomeres [205].

##### Long Non-Coding RNAs

PSCs express a characteristic set of lncRNAs [290,291], which is actively involved in the maintenance of the pluripotency status [274,275,292]. More than one thousand pluripotency-associated lncRNAs act as “regulators of regulators” by interacting with the PGRN, or operate as “pivots of pluripotency” acting as a barrier against differentiation [292,293]. Indeed, lncRNAs silencing is associated to pluripotency loss and the beginning of differentiation [294]. For the maintenance of pluripotency in the mouse, lncRNA-1592 and lncRNA-1552 regulate and are regulated by the pluripotency transcription factors (including Oct-4, Nanog and Klf4) in a positive feedback loop, whereas the lncRNAs GOMAFU and AK141205 are direct targets of Oct-4 and Nanog, respectively [294]. In addition, pluripotency is sustained by lncRNA TUNA/MEGAMIND, which, by forming a complex with Nucleolin and other ribonucleoproteins, bind *Nanog*, *Sox2* and *Fgf4* promoters [295]. 

In hESCs, the pluripotency status is regulated also by the lncRNA RoR (Regulator of reprogramming), which acts in the cytoplasm by endogenous competition with miR-145, the latter being able to block pluripotency transcription factors translation [296]. LncRNAs ES1, ES2 and ES3, which directly associate with SUZ12 (Polycomb Group (PcG)) and SOX2 [297], together with lincRNA-p21, a nuclear noncoding transcript repressor of the p53-dependent transcriptional cascade [298,299], contribute to the pluripotency maintenance. In general, depletion of one or more of these lncRNAs results in pluripotency perturbation (decreased expression level of pluripotency factors) and inhibition of the in vitro self-renewal capacity and differentiation induction [272]. 

Several pluripotency-associated lncRNAs are also involved in the modulation of the activity of epigenetic regulators and guide chromatin remodelers and histone modifiers, such as Polycomb and Tritorax complexes [300], acting as a scaffold by regulating nucleosomal structures or by modulating post-translational modifications on histone tails. For instance, in both mouse and human PSCs, a number of lncRNAs (e.g., Meg3, ANRIL, RIAN and MIRG and other lncRNAs from the imprinted Dlk1-Dio3 locus) temper PRC2 interaction with its cofactors via Jumonji and AT-Rich Interaction Domain Containing 2 (JARID2) [301,302]. Meg3 and other lncRNAs work as scaffolds, favoring the interaction between JARID2 and PRC2 core components. In addition, these lncRNAs may also guide the initial recruitment of PRC2/JARID2 at specific target sites via RNA-DNA base-pairing. LncRNAs interact with WD repeat-containing protein 5 and with other Tritorax-related factors (including mammalian MLL complexes), positively controlling transcription by promoting H3K4me3 deposition.

## 5. Conclusions

In the last few decades, studies on mouse and human peri-implantation embryos were aimed at the identification of the main genetic and epigenetic factors involved in determining the potency of cells at the beginning of development and its transition towards a pluripotency status while reaching the peri-implantation stages. While the knowledge of the mouse embryo peri-implantation developmental processes has reached a fine characterization of the cellular and molecular events that guide the cell potency transition towards the achievement of pluripotency, technical and ethical limitations have delayed a similar understanding of human peri-implantation embryos events. Nevertheless, the knowledge so far gathered has evidenced several differences between these two species, suggesting that the development processes may be differentially regulated in Mammals. The differences mainly concern the timing of the cellular events and molecular players leading to the achievement of the first and the second lineage specification, from the zygote to the peri-implantation embryo, as well as the epigenetic machinery that regulate and accompany the achievement of a pluripotency status. The opportunity to derive and maintain in vitro different cell lines from both mouse and human embryos has allowed a significant improvement in the comprehension of the determinants involved in the establishment of the cell identity and in the maintenance and/or progression towards the different states of the cell potency. In particular, for the mouse, changes in the pluripotency states have been captured and transferred in vitro. However, the pluripotency status of these cell lines represents a static picture of the cell potency continuum in vivo and, in addition, it can be significantly influenced by the culture conditions used for their derivation and maintenance. Nevertheless, each single static picture starts to take its specific place in a more complex scenario. Indeed, based on the moment of isolation, the genes expressed, the pathways activated or repressed, the epigenetic signature and the wide spectra of regulating molecules (e.g., non-coding RNAs) define a distinguishable and identifiable naïve, rosette, formative and primed pluripotency status. 

Future studies, using new technologies (e.g., high resolution and time-lapse imaging, single cell next generation sequencing, single cell proteomic analysis, advanced cell images and computational models) and 3D culture techniques, which allow to mimic and reproduce in vitro the whole peri-implantation period of development, are needed to precisely define how these pluripotency states functionally and transcriptionally relate to one another and whether they reflect bona fide counterparts in embryos. 

## Figures and Tables

**Figure 2 cells-10-02049-f002:**
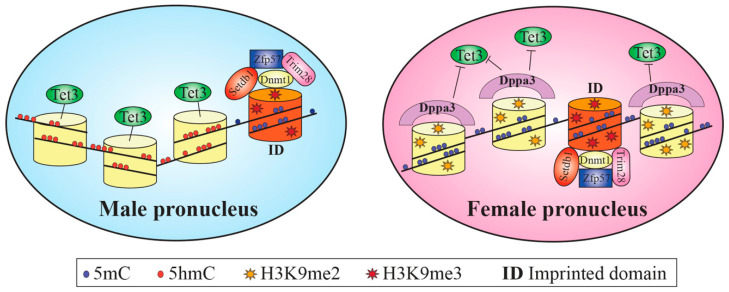
Schematic representation of epigenetic reprogramming occurring in the male and female pronuclei. The male pronucleus undergoes an active, almost global, erasure of DNA methylation, mainly achieved through the Tet3 enzyme, with the exception of imprinted loci. The presence of high levels of histone H3K9me2 together with the presence of Dppa3, which reduces the affinity of Tet3, protects the female genome from active demethylation. Trim28, Stedb1 and Zfp57 form a complex that protects imprinted domains; Dnmt1 allows the maintenance of cytosine methylation during DNA replication.

**Figure 3 cells-10-02049-f003:**
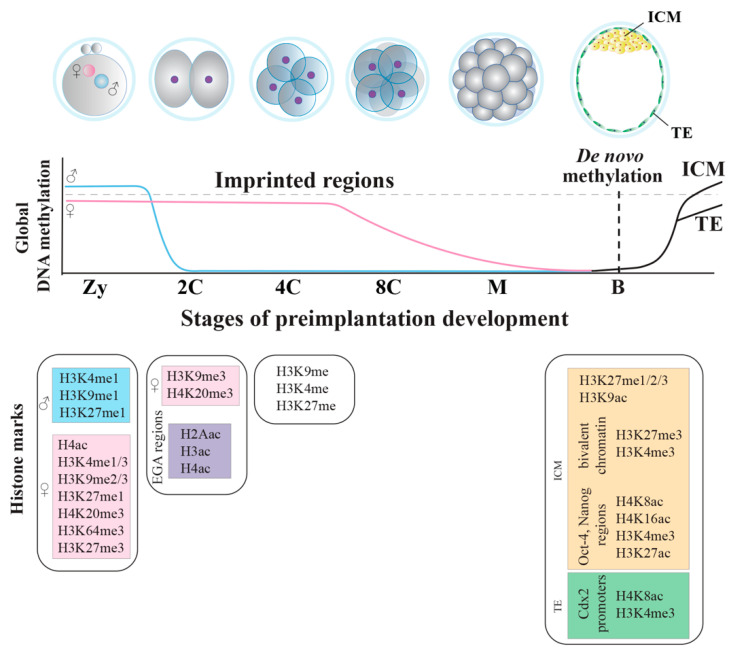
Methylation dynamics during mouse embryonic preimplantation development. After fertilization, the male genome (blue) undergoes fast demethylation, whereas the female genome (pink) undergoes a slow passive DNA demethylation. De novo methylation occurs at the blastocyst stage, with a differential methylation pattern between the ICM and the TE. Specific histone marks are present in the male and female genomes and in the embryo at different pre-implantation stages. Zy, zygote. C, cell. M, morula. B, blastocyst. ICM, inner cell mass. TE, trophectoderm.

**Figure 4 cells-10-02049-f004:**
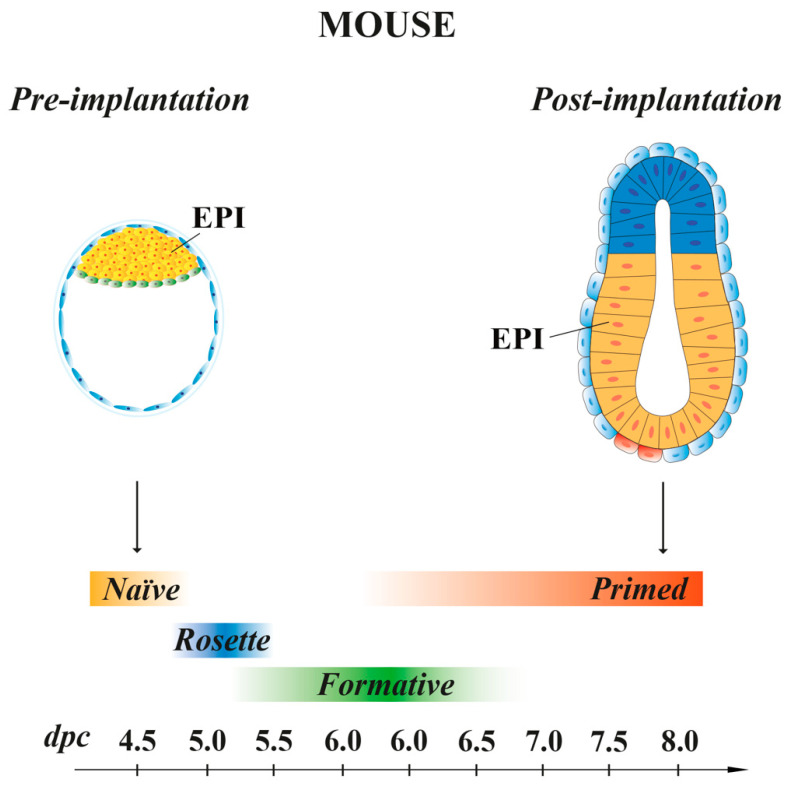
Pluripotency status of stem cells (SCs) derived from 4.5–8 days *post coitum* (dpc) embryos. ESCs derived from the mouse blastocyst have a naïve pluripotency; EpiSCs derived from post-implantation embryos have a primed pluripotency. SCs derived from either 5–6.5 dpc mouse embryos or naïve ESCs possess rosette or formative pluripotency, intermediate between the naïve and primed pluripotency. EPI, epiblast.

**Figure 5 cells-10-02049-f005:**
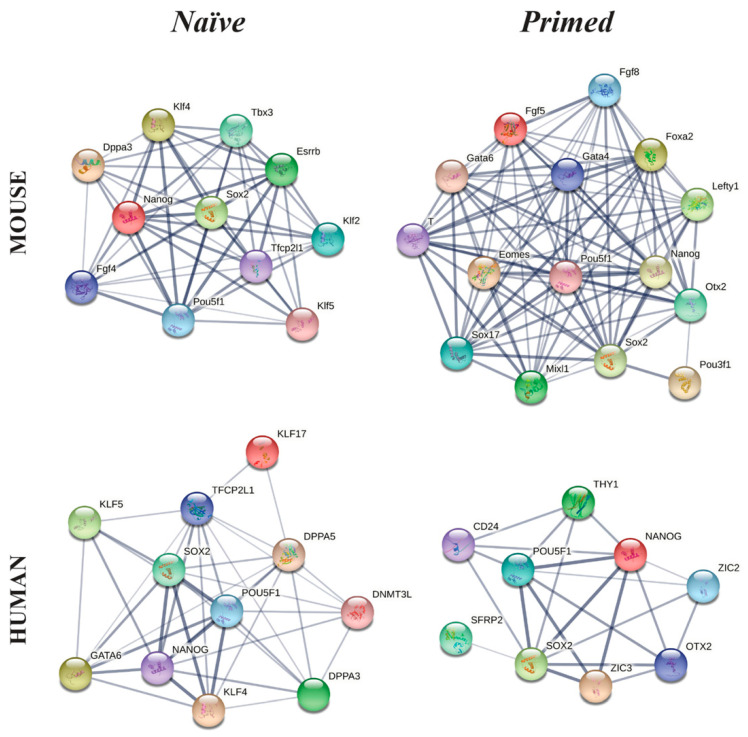
Functional mouse and human protein association networks of the naïve or primed pluripotency status, obtained using String (https://string-db.org/ accessed on 1 November 2020).

## Data Availability

Not applicable.

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
