# Peer review of "Building Pluripotency Identity in the Early Embryo and Derived Stem Cells"

_cells, 2021, doi:10.3390/cells10082049_

Round 1
Reviewer 1 Report
This review addresses the molecular identity of pluripotency in embryos and stem cells. It aims to provide insight into how the totipotent state of the zygote transitions into the naive and primed pluripotent states of pre- and postimplantation epiblast, and how these pluripotent states are captured in pluripotent stem cells. To this aim, the authors describe certain transcriptome and epigenome features in preimplantation embryos and in pluripotent stem cells, with an emphasis on non-coding RNA molecules. I have a number of serious concerns with this manuscript. The authors seem to have no clear concept of what they want to convey to their audience but list an arbitrary collection of research findings. They provide little context and make no effort to interpret these findings or indicate their relevance or how they are connected or contribute to pluripotency. It is unclear what insights about the molecular identity of pluripotency we are supposed to gain from this manuscript and why anybody would want to read it. Furthermore, the manuscript suffers from many factual errors and the authors display a limited grasp of embryology and concepts of stem cells and pluripotency. Frequently they make assertions about embryos based on studies that were performed only with stem cells. Later on, they argue that these stem cells represent the embryonic tissues because of these similarities that were in fact never demonstrated. I felt at times that I was correcting a draft of a student’s report. Below I list a number of such issues I found, not complete. The authors should carefully reconsider how they approach writing a review. I would advise picking a much more limited topic in which they are knowledgeable and can aggregate and interpret recent findings to come to an original contribution. I regret to say that the current effort is in my opinion not suitable for publication.
Specific issues:
I’m unfamiliar with the term ipoblast (line 46), which seems to be Italian for hypoblast.
L47: it is the ICM, not the epiblast, that gives rise to both embryonic and extraembryonic tissues (hypoblast and epiblast). In fact, the epiblast also gives rise to extraembryonic mesoderm. Either way, the current formulation is wrong and the authors seem not to understand the difference between ICM and epiblast.
L55: mouse ‘post-implantation blastocysts’ don’t exist.
L64: EpiSCs represent a postimplantation stage and were not originally derived from preimplantation embryos.
L63: ‘In this Review we will describe the stemness molecular identity of the preimplantation embryo, from zygote to blastocyst, and of the derived pluripotent ESCs and EpiSCs.’ This is an important sentence because it describes the focus of the review. Unfortunately, it is overly vague and grammatically incorrect. Can the authors explain what specifically they mean with ‘stemness molecular identity’ and how they envision the presence of ‘stemness’ in the preimplantation embryo?
Many citations to old review articles, like 22. Original research articles can of course be very old, but there is no good reason to cite review articles decades old if ones incorporating more recent findings are readily available.
L78: Maternal clearance enables EGA, which is required for degradation of maternal transcripts, which is the same as maternal clearance, is it not? This is contradictory. What is the difference between maternal clearance and degradation of maternal transcripts?
L80: TBP and TFIID are not transcription factors.
Figure 1 summarizes a number of embryonic events but is only referred to regarding the timing of embryonic genome activation. The potency timeline ignores the extraembryonic tissues. Totipotency is not substituted by pluripotency but distributed over the daughter cells, some of which retain pluripotency. It also confuses potency with the presence of stem cells. However, also non-stem cells can have a differentiation potential, and totipotent cells are not stem cells.
L98: Zscan4 is the best characterized 2-cell specific gene but known for its role in ES cells? Then it is not a 2-cell specific gene.
L104: what is this cell compaction that occurs between 8-cell and morula stages? And what marked changes in expression profile determine that? What does it have to do with embryo compaction, or is it the same? Embryo compaction occurs later.
L114-118: Oct4, Nanog and Sox2 represent the functional core of the pluripotency network, while Oct4 and Sox2 represent its heart? What is the difference between functional core and heart?
L126: ‘… prominent hubs whose connections concur to maintain the pluripotency state of the ICM in vivo’. This has only been demonstrated in ES cells, not in embryos.
L128-130: This single sentence refers to 15 papers with only a vague notion of their general topic and no idea of what they demonstrate. The purpose of a review is to summarize and place in context the key findings of the research literature.
L135-136: This is not demonstrated by the cited papers.
L137: ‘Erasure of the gametic genomes is necessary to shepherd the totipotency state and to support EGA and the early developmental phases.’ What does ‘shepherd the totipotency state’ mean, and where is all this demonstrated?
L146: This has only been demonstrated in ES cells, not in embryos.
The reference list contains multiple duplications. Authors seem to create a new citation whenever a paper is cited.
L171: This is not demonstrated by the cited paper.
L175: This is not demonstrated by the cited paper (concerns only mouse).
Section 2.1.1. lists many histone modifications without providing any information of their roles and functions, which makes the significance of this information unclear.
Figure 2: male/female or paternal/maternal pronucleus? These mean different things.
Line165-166: I have never heard of cytosine trimethylation.
L183: Here the authors claim that de novo methylation occurs between the morula and the blastocyst stage in both mouse and human embryos, without providing any citation. In fact, de novo methylation starts only after the blastocyst stage, see e.g. Smith et al 2012 Nature 484, 339.
L187: This is not demonstrated by the cited papers.
L193: Not demonstrated by the cited paper (97).
L197-198: The cited papers show that bivalency is in fact absent in the preimplantation epiblast and therefore not characteristic at all for pluripotency in vivo.
L202: Not demonstrated by the cited paper, which concerns ES cells rather than ICM.
L220: morula blastomeres don’t exist
2.1.2. X-reactivation is forgotten.
Figure 4: less pluripotency in E5.5 embryo?
L305-306: mixup of mouse and human naïve genes.
L304-329: Their concept of heterogeneity in ES cell cultures and the self-renewal signals that maintain the equilibrium is outdated.
L324: interconversion of Epi and PrE precursors has been extensively studied.
L329: ‘genes that regulate pluripotency in vitro are not necessarily required for early EPI development in vivo.’ It’s the other way around. Examples?
L330-346: concept of heterogeneity in EpiSCs ignored.
L345: wrong citation for naïve-like EpiSCs.
L350-354: concept of formative pluripotency misrepresented and outdated.
L385-390: Multiple studies over the past 5 years demonstrated a dynamic equilibrium in DNA methylation in pluripotent cells, something that is ignored here.
Overall, section 3.1.1 sums up many observations with very little context or interpretation.
Reviewer 2 Report
GENERAL COMMENT
In this manuscript, Rebuzzini et al. review the current knowledge on mouse and human early embryonic development and the corresponding pluripotent stem cells. They address this subject in a comprehensive manner, describing diverse processes at multiple cellular levels, covering the epigenetic landscapes, gene expression profiles, non-coding RNAs and transcription factor interaction networks. Throughout the manuscript, they compare the mouse and the human embryos and PSCs, showing similarities and differences between the two species. Thus, they provide relevant information highlighting assets and limitations of the mouse model, which gives elements to support the necessity for human embryo research.
While the subject is globally well addressed, more rigor on terminology would strengthen it, especially regarding the human part. Although the manuscript is mainly based on mouse data, the authors should provide more examples of molecular actors involved in the described events occurring in human. They also consider “cell potency” in a global manner, describing the “cell potency loss” through developmental progression, e.g through the transition from totipotency to pluripotency. While the idea is quite straightforward, this notion remains too vague and they should rather describe how cell potency is progressively restricted through development. Finally, the review would benefit from more connections between the epigenetic landscape, transcriptional regulation and the pluripotency associated transcription factor networks.
MAJOR
- In the title “The Molecular Identity of Pluripotency in Preimplantation Embryos and in Their Derived Stem Cells”, you should replace “preimplantation” with “peri-implantation”. Indeed, you describe some processes and cellular states, like EpiSCs and primed hESCs, that are found after implantation, and today it is not relevant to limit the analysis to the pre-implantation period, as we can easily access mouse embryos at any time of development and human embryos until day 14. More globally, you should include in your review the recent findings on human peri-implantation embryos up to day 14. This would strengthen your message and increase the significance of this review, by giving a complete, up-to-date picture of pluripotency throughout this developmental period in both species (Xiang et al., 2020; Zhou et al., 2019).
- Line 29: “The term “totipotency” defines the capability of a cell to generate all differentiated cell types of a new developing organism, or even the ability to develop into an entire organism [2,3].” This statement is not accurate, as the term "totipotency" exclusively defines the ability of a cell to develop into an entire organism. Alternatively, the term "plenipotency" has been proposed to describe a cell that is able to generate all differentiated cell types, including extra-embryonic annexes, but has lost the capacity to produce a new organism (Condic, 2014).
- Line 40: “In human, when and how blastomeres loose totipotency is still unknown [11].” This statement is probably a bit too vague. It would be better to clarify is the intent is to explain that cells at the blastocyst stage are still plenipotent? Or else, when is clearly after compaction. For the “how” part, Gerri et al (Nature 2020) bring some important understanding of that development period.
- Line 47: “ICM and EPI cells display a pluripotent status, the latter cells giving rise to both embryonic and extraembryonic tissues”. This statement lacks clarity. ICM cells cannot be defined as pluripotent. In the mouse, it has been shown that ICM is initially made of common progenitors of EPI and PrE, characterized by the co-expression of Nanog and Gata6 (Saiz et al., 2016). Only EPI cells that downregulate Gata6 can be considered pluripotent. In human, no ICM specific molecular signature has been identified so far, and only the EPI can be considered pluripotent (e.g: core pluripotency associated transcription factor network, differentiation potential). Moreover, although some EPI cells indeed produce at later stages extraembryonic amnion cells, it is somewhat misleading to associate pluripotency with this event, as pluripotency typically entails the potential to produce the three germ layers and primordial germ cells.
- Suggested text changes :
Line 56: “These different types of PSCs are characterized by distinguishable molecular and functional signatures and differentiation potential [20,21].” One cannot say that ESCs and EpiSCs have distinct differentiation potential, as they are both able to produce the three germ layers and primordial germ cells (Huang et al., 2012). However, mESCs residing in the naïve state require capacitation prior to engage differentiation towards the three germ layers and PGCs (Mulas et al., 2017).
Line 70 : “Although mouse and human pre-implantation development are similar in their cytological events”. Developmental timings are quite different, see Gerri et al. 2020. Replace with “show similarities” or other sentence tempering the statement.
Line 100 : “Only 40% of human embryonic-expressed genes are shared with the mouse and many of them have unknown function”. Please add a reference.
Line 104-107 : “After EGA, molecular differences among blastomeres arise. Cell compaction, that in mouse occurs between the 8-cell and morula stages, is determined by marked changes in the embryonic expression profile [40]. E-cadherin and several cytoskeleton and cell adhesion/junction-related genes are induced and expressed at high levels at the morula stage, permitting embryo compaction and the first specification processes [41]. »
More precision would be nice: which changes, events, proteins? You can detail HIPPO pathway here. Moreover, compaction occurs in mouse at 8 cells, before the morula stage. This is the increase of polarization proteins concomitantly to compaction that allow the embryo to reach the morula stage with inner/outer cells. Asymetric polarization induce YAP translocation in outer cells nucleus thus triggering TE specification of outer cells. See Chazaud and Yamanaka 2006 and Gerri et al. 2020.
- Line 169: « The maintenance of maternal methylation profile is more prominent in human than in the mouse, indicating, in the former species, a more active role of DNMT1 in the maintenance of DNA methylation ». I could not find any element corroborating this statement in the reference you cite (Hanna et al., 2018). This difference between the two species could be simply due to the global delay in development of the human embryo relative to the mouse.
- Line 224: “Upon blastocyst implantation, mouse EPI cells undergo random XCI”. You should also mention and describe the process of X chromosome reactivation (XCR) that occurs in mouse EPI cells prior to the second wave of random XCI.
- Line 227: “Differently from the mouse, in human preimplantation embryos, the precise mechanism of X dosage compensation remains still under discussion [113] and the X dampening [114] or the X inactivation [115] models have been proposed”. Observation was made that biallelic expression of X-linked genes progressively decreases in human female pre-implantation embryos, which is followed by monoallelic expression after implantation (Petropoulos et al., 2016; Zhou et al., 2019). Although it is actually debated, one current model is as follows: X chromosome dampening (XCD) is happening shortly after EGA, resulting in a decrease in biallelic expression through preimplantation, then random X chromosome inactivation (XCI) occurs by the time of implantation for dosage balance between females and males, which is followed by an upregulation of active X-linked genes for dosage balance with autosomal genes (Petropoulos et al., 2016; Sahakyan et al., 2017; Zhou et al., 2019).
- Line 272: The part regarding Long-Non-Coding RNAs is too sparse. You should find other examples and describe their function during peri-implantation development. For instance, you could describe in more details here the mechanism by which the lncRNAs XIST and XACT are driving XCI, to link it to other parts of the manuscript.
- Line 301: « efforts are made to convert primed hESCs into the naïve state in vitro, or to isolate them in a naïve-like pluripotent condition [135-139] ». This statement is misleading, you should rephrase. Indeed, since 2014, we can efficiently convert primed hESCs into naïve hESCs (Takashima et al., 2014), and since 2016, we can derive these cells directly from human ICM (Guo et al., 2016). Naïve hESCs long-term self-renew and can be efficiently expanded in vitro.
- Line 301: “Naïve and primed pluripotency states represent a continuum during development”. In both mouse and human, alternative pluripotency states have been reported, like EPS (extended pluripotent stem cells, (Yang et al., 2017b)) and EPSC (expanded potential stem cells, (Gao et al., 2019; Yang et al., 2017a)). You could describe these states and discuss their potency and relation with the embryo.
- Line 323: “In vivo, the ICM is also composed by a mix of EPI and PrE-precursors”. These cells are thought to be bipotent progenitors of both EPI and PrE (Saiz et al., 2016). Moreover, this statement is not relevant here, as it is not related with the preceding description of heterogeneity among ESCs which does not include PrE cells.
- The conclusion part is rather short. It would be nice to include an integrated perspective, highlighting the connections between the processes that you describe throughout the review. For example, you could speculate on mouse vs human, or embryo vs cellular models.
Proposed minor modifications:
Line 15: replace “i.e” with “e.g”
Line 17: replace “the first differentiation events” with “the first specification events”
Line 19: replace “the passage from a totipotent cell” with “the transition from a totipotent cell”
Line 41 (onwards): “blastomere compaction (E 3-3.5 days post coitum, dpc, in the mouse; E 5.0 dpc in human)”. You should use days post-fertilization (dpf) for human embryo staging as it is more accurate. In this case, compaction lasts until day 4.
Line 44: “While TE cells have limited potency”. Replace “limited” with “restricted”.
Line 46 (onwards): “the ipoblast (IPO, or primitive endoderm, PrE)”. Although the term hypoblast is still used sometimes, you might want to use primitive endoderm (PrE) to avoid any confusion with other tissues.
Line 136: replace “genomes” with “epigenomes”
Line 226: You should add “after which the inactive X is clonally inherited to the cell progeny” at the end of this statement.
Line 328: « Indeed, genes that regulate pluripotency in vitro are not necessarily required for early EPI development in vivo ». You could include a reference here.
Line 379: replace “relevant” with “significant”
Line 472: « In general, depletion of one or more of these lncRNAs results in pluripotency perturbation (decreased expression level of pluripotency factors) and inhibition of the in vitro self-renewal capacity and differentiation induction. » You should include a reference here.
It might be better to have TE and male pronuclei in different colors in figure 3, to avoid confusion. It seems relevant to indicate EGA. Also, it is not clear whether the schematic is for human or mouse (in the mouse at the morula stage, inner cells (predictive ICM) vs outer cells (predictive TE) are already specified although not yet committed).
Figure 4 is acceptable, but you should enhance contrast to better distinguish between pre- and post-implantation cells. You should also remove PrE (« IPO ») from the legend, as it does not relevant here.
Figure 5: It would be nice to add human pluripotency networks comparison.
Condic, M.L. (2014). Totipotency: what it is and what it is not. Stem Cells Dev 23, 796-812.
Gao, X., Nowak-Imialek, M., Chen, X., Chen, D., Herrmann, D., Ruan, D., Chen, A.C.H., Eckersley-Maslin, M.A., Ahmad, S., Lee, Y.L., et al. (2019). Establishment of porcine and human expanded potential stem cells. Nature cell biology 21, 687-699.
Guo, G., von Meyenn, F., Santos, F., Chen, Y., Reik, W., Bertone, P., Smith, A., and Nichols, J. (2016). Naive Pluripotent Stem Cells Derived Directly from Isolated Cells of the Human Inner Cell Mass. Stem cell reports 6, 437-446.
Hanna, C.W., Taudt, A., Huang, J., Gahurova, L., Kranz, A., Andrews, S., Dean, W., Stewart, A.F., Colomé-Tatché, M., and Kelsey, G. (2018). MLL2 conveys transcription-independent H3K4 trimethylation in oocytes. Nature Structural & Molecular Biology 25, 73-82.
Huang, Y., Osorno, R., Tsakiridis, A., and Wilson, V. (2012). In Vivo differentiation potential of epiblast stem cells revealed by chimeric embryo formation. Cell reports 2, 1571-1578.
Mulas, C., Kalkan, T., and Smith, A. (2017). NODAL Secures Pluripotency upon Embryonic Stem Cell Progression from the Ground State. Stem cell reports 9, 77-91.
Petropoulos, S., Edsgärd, D., Reinius, B., Deng, Q., Panula, Sarita P., Codeluppi, S., Plaza Reyes, A., Linnarsson, S., Sandberg, R., and Lanner, F. (2016). Single-Cell RNA-Seq Reveals Lineage and X Chromosome Dynamics in Human Preimplantation Embryos. Cell 165, 1012-1026.
Sahakyan, A., Plath, K., and Rougeulle, C. (2017). Regulation of X-chromosome dosage compensation in human: mechanisms and model systems. Philosophical transactions of the Royal Society of London Series B, Biological sciences 372.
Saiz, N., Williams, K.M., Seshan, V.E., and Hadjantonakis, A.K. (2016). Asynchronous fate decisions by single cells collectively ensure consistent lineage composition in the mouse blastocyst. Nature communications 7, 13463.
Takashima, Y., Guo, G., Loos, R., Nichols, J., Ficz, G., Krueger, F., Oxley, D., Santos, F., Clarke, J., Mansfield, W., et al. (2014). Resetting transcription factor control circuitry toward ground-state pluripotency in human. Cell 158, 1254-1269.
Xiang, L., Yin, Y., Zheng, Y., Ma, Y., Li, Y., Zhao, Z., Guo, J., Ai, Z., Niu, Y., Duan, K., et al. (2020). A developmental landscape of 3D-cultured human pre-gastrulation embryos. Nature 577, 537-542.
Yang, J., Ryan, D.J., Wang, W., Tsang, J.C.-H., Lan, G., Masaki, H., Gao, X., Antunes, L., Yu, Y., Zhu, Z., et al. (2017a). Establishment of mouse expanded potential stem cells. Nature 550, 393-397.
Yang, Y., Liu, B., Xu, J., Wang, J., Wu, J., Shi, C., Xu, Y., Dong, J., Wang, C., Lai, W., et al. (2017b). Derivation of Pluripotent Stem Cells with In Vivo Embryonic and Extraembryonic Potency. Cell 169, 243-257.e225.
Zhou, F., Wang, R., Yuan, P., Ren, Y., Mao, Y., Li, R., Lian, Y., Li, J., Wen, L., Yan, L., et al. (2019). Reconstituting the transcriptome and DNA methylome landscapes of human implantation. Nature 572, 660-664.
Reviewer 3 Report
This review article provides a comprehensive and chronologically landscape of the gene expression, epigenetics, miRNA, X chromosome inactivation, and non-coding RNA in the implanting embryo in vivo and pluripotent stem cells in vitro between mouse and human. This paper has high value as a review in the field of embryology and its related fields and might be accepted with no revision.
Reviewer 4 Report
In this review Rebuzzini et al discuss the transcriptional and epigenetic mechanisms that regulate totipotency and pluripotency, in mouse and human embryos and mouse and human embryonic stem cells. The review is clearly organised, and the topic is relevant for the community. The figures are clear and informative. However, I have identified a number of major mistakes in the review along with literature omissions as outlined below:
1. Line 36: the authors state that in the mouse each blastomere from a 2-cell embryo is totipotent and capable of developing into a complete organism. This statement is not entirely correct. Casser et al, 2017, Scientific Reports separated the blastomeres of 1,252 two-cell stage embryos and found that totipotency is not equally inherited by the two blastomeres. The authors should clearly explain the different experimental evidence pro/against the two hypotheses.
2. Line 41: blastomere compaction does not happen at E3.5 in the mouse embryo and at E5.0 in human embryos. It happens at E2.5 in the mouse and at E4 in the human.
3. Line 74: the authors claim that there is a single wave of EGA in human embryos but there may also be a minor one (Taylor et al, 1997; Vassena et al, 2011).
4. Figure 1: the drawings of the E5.5 mouse embryo and E10 human embryo are exactly the same as the drawings shown in Shahbazi et al, Science, 2019. The authors should acknowledge this in the legend.
5. Line 104: compaction in the mouse embryo does not occur between the 8-cell and morula stages, it always occurs at the 8-cell stage.
6. Line 280: the authors have a section dedicated to the role of lncRNAs during mouse embryo development, and yet do not discuss the relevant papers, just state “lncRNAs are scarcely documented”. For example, Wang et al, Cell, 2018 explored the role of LincGET lncRNA during mouse embryo development.
7. Line 289: mouse ESCs retain the transcriptional profile of E4.5 (not E3.5). This was shown by Boroviak et al, 2014.
8. Line 326: the authors state that during embryo diapause there is an interconversion of EPI and PrE progenitors. I am not aware of such a phenomenon been described in the literature. They cite Plusa et al, 2008 to support this claim, but Plusa et al do not mention diapause embryos.
9. Line 343: Fgf5 is not a naïve marker.
10. Line 344: the authors talk about the distal enhancer of Oct-4, but without more context the reader will not understand what this means. Moreover, they cite reference 149, which is about reprogramming in human cells, and not relevant to what is been discussed.
11. The authors discuss the role of miRNAs in totipotency and pluripotency but they do not discuss work that has shown exposure to miR-203 extends the potency of pluripotent stem cells (Salazar-Roa et al, 2020). Moreover, the section regarding totipotency would benefit from a description of the different stem cell cultures that have been proposed to confer totipotent-like features to embryonic stem cells.
12. Some of the sections in the review provide a detailed enumeration of genes, but fail to convey a critical insight. I think the authors should provide a critical assessment of the current literature. This could be done at the end of every section, or at the end of the review (future perspectives section).
Minor comments:
- Line 46: Ipoblast is misspelled, it should be written hypoblast.
- The formatting of the references is wrong in various places
Round 2
Reviewer 4 Report
The authors have satisfactorily addressed all of my comments. The manuscript has been greatly improved. I think it should be accepted for publication.